# Connectivity guided theta burst transcranial magnetic stimulation versus repetitive transcranial magnetic stimulation for treatment-resistant moderate to severe depression: study protocol for a randomised double-blind controlled trial (BRIGhTMIND)

Richard Morriss ,[1] Lucy Webster,[2] Mohamed Abdelghani,[3] Dorothee P Auer,[4,5] Shaun Barber,[6] Peter Bates,[7] Andrew Blamire,[8] Paul M Briley,[9] Cassandra Brookes,[10] Sarina Iwabuchi,[9] Marilyn James,[11] Catherine Kaylor-Hughes,[9] Sudheer Lankappa,[2] Peter Liddle,[9] Hamish McAllister-Williams,[8] Alex O'Neill-Kerr,[12] Stefan Pszczolkowski Parraguez,[13] Ana Suazo Di Paola,[10] Louise Thomson,[1] Yvette Walters,[10] BRIGhTMIND study team

For numbered affiliations see end of article.

**Correspondence to**
Professor Richard Morriss;
richard.morriss@nottingham.ac.uk

## ABSTRACT

**Introduction** The BRIGhTMIND study aims to determine the clinical effectiveness, cost-effectiveness and mechanism of action of connectivity guided intermittent theta burst stimulation (cgiTBS) versus standard repetitive transcranial magnetic stimulation (rTMS) in adults with moderate to severe treatment resistant depression.

**Methods and analysis** The study is a randomised double-blind controlled trial with 1:1 allocation to either 20 sessions of (1) cgiTBS or (2) neuronavigated rTMS not using connectivity guidance. A total of 368 eligible participants with a diagnosis of current unipolar major depressive disorder that is both treatment resistant (defined as scoring 2 or more on the Massachusetts General Hospital Staging Score) and moderate to severe (scoring ≥16 on the 17-item Hamilton Depression Rating Scale (HDRS-17)), will be recruited from primary and secondary care settings at four treatment centres in the UK. The primary outcome is depression response at 16 weeks (50% or greater reduction in HDRS-17 score from baseline). Secondary outcomes include assessments of self-rated depression, anxiety, psychosocial functioning, cognition and quality of life at 8, 16 and 26 weeks postrandomisation. Cost-effectiveness, patient acceptability, safety, mechanism of action and predictors of response will also be examined.

**Ethics and dissemination** Ethical approval was granted by East Midlands Leicester Central Research Ethics Committee (ref: 18/EM/0232) on 30 August 2018. The results of the study will be published in relevant peer-reviewed journals, and then through professional and public conferences and media. Further publications will explore patient experience, moderators and mediators of outcome and mechanism of action.

**Trial registration number** ISRCTN19674644

### Strengths and limitations of this study

► Large multicentre randomised controlled trial of connectivity guided theta burst stimulation intermittent (TBS) versus standard repetitive transcranial magnetic stimulation (rTMS).
► Carefully characterised population of patients with treatment-resistant moderate to severe unipolar major depression with broad inclusion and exclusion criteria to enhance generalisability to clinical care.
► Explores efficacy, sustainability of treatment response, acceptability, cost-effectiveness cognitive function, mechanism of action using MRI and spectroscopy, and clinical and MRI predictors of response in the same sample.
► Unable to completely blind sound and sensation of the treatments being tested.
► Both nature of magnetic stimulation (TBS vs rTMS) and stimulation site identification vary between treatment arms.

## INTRODUCTION

The lifetime prevalence of major depressive disorder (MDD) is approximately 13% of the general population,[1] and it is the most disabling health condition in terms of years lived with disability.[2] Antidepressants and

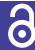

psychotherapies are effective in treating MDD. However, at least 33% of patients in specialist care,[3] and 22% in general primary care,[4] fail to respond to adequate trials of at least two antidepressants. Therefore, it is essential to find alternative treatments for such 'treatment-resistant depression' (TRD).

Neuromodulation techniques that attempt to directly modulate the function of targeted brain regions, such as electroconvulsive therapy (ECT),[5] and repetitive transcranial magnetic stimulation (rTMS),[6] can be effective treatments for MDD. rTMS uses powerful magnetic pulses that can be focused usually on the left dorsolateral prefrontal cortex (DLPFC) to produce changes in brain activity if it is given as a course of treatments. Unlike ECT, rTMS does not require anaesthesia nor produce potential cognitive deficits, and is cost-effective in TRD compared with current standard care.[7]

A recent meta-analysis in TRD demonstrated the acute effectiveness of rTMS when compared with sham TMS, showing significant reductions in depression symptoms, and increases in response and remission and response rates, after treatment.[6] However, the beneficial effects of rTMS on mood in TRD may be relatively short, lasting only 1–3 months.[6 8]

Theta burst stimulation (TBS) is an alternative patterned form of TMS, which employs high frequency stimulation, with each treatment administration requiring less time. Unlike rTMS, TBS uses bursts of magnetic pulses that mimic endogenous theta rhythms and it is associated with cortical long-term potentiation that may induce plasticity in more distal brain areas, such as the hippocampus, potentially leading to a higher proportion of responders and a longer duration of effect in TRD.[9] A meta-analysis of five randomised controlled trials (RCTs) in 221 patients with MDD,[10] found unilateral intermittent TBS (iTBS) applied to the left DLPFC and consecutive iTBS applied to the left DLPFC and then continuous TBS applied to the right DLPFC to be robustly effective versus sham TBS on depression symptoms and response. One RCT demonstrating 83% of TRD patients given unilateral iTBS maintained their response at 14 weeks unlike sham and bilateral TBS.[11] A large RCT of 414 TRD patients comparing iTBS versus rTMS to the left DLPFC showed a non-inferior reduction in depression symptoms after 4–6 weeks of treatment and at 1, 4 and 12 weeks after treatment.[12] Thus, TBS requires a shorter duration of administration than rTMS without compromising clinical effectiveness. Further studies are required to maximise efficacy and durability of effect of TBS, identify its mechanisms of action and predictors of response.

Brain connectivity changes as detected by task-free, resting state functional MRI (rsfMRI) may individualise neurostimulation therapy of MDD,[13] with the insula suggested as a target for neuromodulation.[14] In depression, an altered network communication exists within and between affective, cognitive control and default mode networks,[15–18] with a disruption of the reciprocal loop between the DLPFC and insula extending to the

sensory regions.[16] In addition, hypometabolism at the right anterior insula (rAI) predicts remission with cognitive–behavioural therapy and similarly, hypermetabolism at the rAI predicts remission with antidepressant treatment in patients with MDD.[19] One biotype of MDD, characterised by strong connectivity between the insula and other regions of the brain, was related to partial treatment response to rTMS of at least 25% improvement on depression symptoms in 82.5% participants.[20]

Therefore, in a small pilot study of 27 TRD patients,[21] individualised left DLPFC targets were determined using Granger causality analysis (GCA) to provide a measure of effective functional connectivity (eFC) seeded from the rAI. Of the 18 participants who completed all follow-up in the two treatment arms (connectivity guided rTMS (cgrTMS) and cgiTBS) higher response rates were found at 3-month follow-up favouring cgiTBS (89%) over cgrTMS (44%), indicating the potential for longer lasting efficacy of cgiTBS.

In addition, rsfMRI and MR spectroscopy (MRS) may advance neuromodulation therapy through mechanistic evaluation of effects and response predictions. iTBS may dampen fronto-insular eFC in responders with TRD.[21] Left DLPFC gamma-aminobutyric acid (GABA) levels are increased significantly in rTMS responders versus non-responders.[22] In addition, DLPFC targeted TMS normalises dysfunctional fronto-limbic networks in TRD, with decreased hyperconnectivity between the subgenual anterior cingulate cortex (ACC), ventromedial and dorsomedial prefrontal cortex and DLPFC.[23] iTBS may induce widespread and longer-term network change but precise anatomical localisation of the target circuitry may be required for maximal efficacy of iTBS to normalise dysfunctional fronto-limbic circuitry. Such changes may increase the proportion of people with TRD who obtain a sustained response compared with neuronavigated rTMS. Thus, this multicentre RCT will examine the clinical and cost-effectiveness of cgiTBS in comparison with neuronavigated rTMS, in treatment-resistant moderate to severe MDD.

## METHODS AND ANALYSIS
### Study design
The study is a multicentre parallel group, double-blind, randomised, controlled trial of the efficacy of cgiTBS versus neuronavigated rTMS without connectivity guidance, in patients with a primary diagnosis of moderate to severe MDD, which is treatment resistant in their current episode (TRD). The study will be carried out at four sites across UK National Health Services (NHS): Nottinghamshire Healthcare Foundation NHS Trust, Northamptonshire Healthcare NHS Foundation Trust, Cumbria, Northumberland, Tyne and Wear NHS Foundation Trust and Camden and Islington NHS Foundation Trust. The study flow chart (figure 1) provides an overview of the study design and procedures.

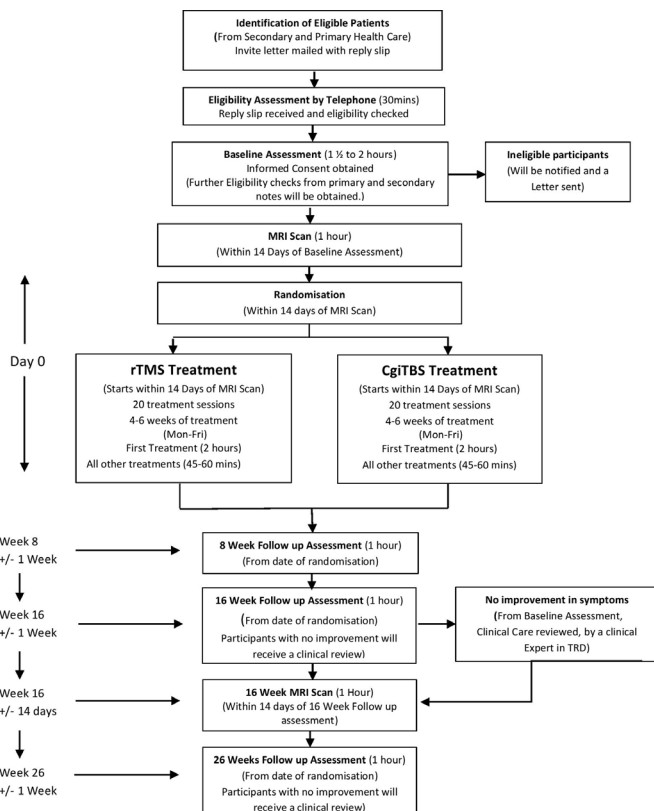

**Figure 1** Study flow chart. cgiTBS, connectivity guided intermittent theta burst stimulation; rTMS, repetitive transcranial magnetic stimulation; TRD, treatment-resistant depression.

## Study population and recruitment

A total of 368 participants will be recruited in total (184 per treatment arm). Recruitment of the first patient started 22 January 2019 but was suspended on 23 April 2020 because of the COVID-19 pandemic lockdown. The study is due to complete on 30 September 2021. Table 1 shows the detailed inclusion/exclusion criteria. Participant identification will occur through primary and secondary care NHS settings across the four sites.

## Interventions

For all treatments, a 70 mm figure-of-eight coil (E-z Cool coil) and a Magstim Horizon Performance Stimulator with StimGuide Navigated TMS Package (Magstim Company, Whitland, UK) will be used. Participants in both treatment arms will receive neuronavigation-guided single coil stimulation for 20 TMS sessions delivered over 4–6 weeks. A total of 3000 pulses will be delivered in each rTMS or cgiTBS session. Each treatment session, irrespective of treatment arm, is designed to last approximately 37.5 min in total for the purposes of blinding the participants and assessors of outcome. There must not be a gap of more than 4 days between treatments in the course of 20. Medication will be unchanged over 16 weeks follow-up unless there is clinical risk of harm.

Individuals assigned to cgiTBS will receive bursts of three pulses (80% motor threshold) at 50 Hz applied at

a frequency of 5 Hz (ie, every 200 ms) for 40 s duration over a site determined from the assessment of maximal strength of effective connectivity between the rAI and the left DLPFC from fMRI and structural MRI. Neuronavigation computes the nearest location for TBS stimulus on the scalp from an individualised head model based on structural MRI and three fiducial points, the nasion, left preauricular and right preauricular sites. If this site causes discomfort to the participant, it is moved by 1 cm in any direction until there is no discomfort. The pulses are repeated for a total of 5 runs with 5 min rest intervals between runs.

Individuals assigned to rTMS will follow the standard US Food and Drug Administration approved protocol. The site of stimulation will be determined using structural and fMRI utilising neuronavigation to compute the F3 electrode site over the left DLPFC for TMS stimulation from the same head model and three fiducial points. Stimulation is at 120% motor threshold with 75×4 s trains of 10 Hz interspersed by 26 s intertrain intervals.

## Primary clinical objective

Our primary hypothesis is that cgiTBS is more efficacious at 16 weeks than standard rTMS in patients with TRD as assessed by the proportion of patients who show a response (50% reduction in depression symptoms from baseline on the 17-item Hamilton Depression Rating Scale (HDRS-17)).

## Secondary clinical objectives

► To explore secondary efficacy outcomes of the mean change in HDRS-17 scores between the two treatment arms and the proportion of patients meeting criteria for remission (HDRS-17 score ≤7) at 16 weeks.
► To explore secondary clinical outcomes of importance to patients and clinicians namely cognition, anxiety, social function and quality of life.
► To examine cost-effectiveness of cgiTBS versus rTMS in a UK NHS population.
► To examine the patient acceptability and patient experience of cgiTBS and rTMS.

## Mechanistic objectives

► To investigate the neural mechanism of efficacy in cgiTBS and rTMS.
► To develop response prediction models from brain biotypes and clinical features.

Protocols for the study will be published separately describing the mechanistic objectives and localisation of sites of stimulation in greater detail.

## Study schedule

Trained researchers under the supervision of clinically trained investigators will obtain informed consent and undertake all assessments. A prescreening questionnaire will be used to telephone screen interested participants first, with potentially eligible participants invited to attend a baseline assessment. Written informed consent

| Table 1 | Inclusion and exclusion criteria |
|---|---|
| **Inclusion criteria** | **Exclusion criteria** |
| ► Adults >18 years<br>► Diagnosis of current MDD (defined according to the DSM-5), that is, treatment-resistant defined as scoring 2 or more on the Massachusetts General Hospital Treatment Resistant Depression staging score (online supplementary appendix 1).<br>► Have a 17-Item Hamilton Depression Rating Scale score of 16 or more (moderate to severe depression).<br>► Capacity to provide informed consent before any trial related activities. | ► History of bipolar disorder (due to risk of mania) or depression secondary to other mental disorder.<br>► Neurological conditions, for example, brain neoplasm, cerebrovascular events, epilepsy, neurodegenerative disorders and prior brain surgery.<br>► Standard contraindications to MRI, that is, irremovable metal objects in and around body, for example, cardiac pacemaker, implanted medication pump, and pregnancy (any doubt resolved by pregnancy test, women of childbearing age taking precautions against pregnancy). This will include other potential complicated factors such as red tattoos which consist of iron on the head, neck and back and claustrophobia (we offer mock scanner testing and training in some sites).<br>► Major unstable medical illness requiring further investigation or treatment.<br>► Change in prescribed medication 2 weeks before baseline assessment.<br>► Prescription of lamotrigine, gabapentin, pregabalin in the 2 weeks prior to baseline assessment but may be used before then.<br>► Daily prescription of benzodiazepine above 5 mg diazepam equivalents, zopiclone above 7.5 mg, zolpidem above 10 mg or Zaleplon above 10 mg. These drugs should not be used intermittently in the 2 weeks before baseline assessment but may be used before then.<br>► Current substance abuse or dependence (defined by DSM-5 criteria).<br>► Prior TMS treatment.<br>► At risk of suicidality.<br>► Potential complicated factors relating to the TMS treatment, that is, hairstyles which would impair magnetic transmission and piercings. (Participants would only be excluded if they chose to not make the changes required to ensure effective treatment.)<br>► Involved with any other clinical trial at the time of consent or 6 months prior.<br>► Unable to read or understand English. |

DSM-5, Diagnostic and Statistical Manual of Mental Disorders, Fifth Edition; MDD, major depressive disorder; TMS, transcranial magnetic stimulation.

will be obtained and study eligibility determined at the baseline assessment.

Information on sociodemographics, medical and psychiatric history including a detailed assessment of treatment resistance will be obtained from case files and primary care notes. Diagnosis will be assessed by the Structured Clinical Interview[24] for the Diagnostic and Statistical Manual of Mental Disorders, Fifth Edition (DSM-5).[25]

Treatment resistance will be measured by the Massachusetts General Hospital (MGH) TRD staging score (adapted for new treatment options—see online supplementary material appendix 1 and table 1).[26]

Depression severity and the primary outcome measure will be assessed using the GRID version of HDRS-17.[27] The HDRS-17 and secondary outcome measures will be assessed in eligible patients at baseline, 8, 16 and 26 weeks postrandomisation (table 2). The primary outcome measure is a binary variable of responder or non-responder at 16 weeks. Individuals observed to have a 50% drop or greater in HDRS-17 from baseline to 16 weeks are defined as responders. In contrast, individuals with <50% reduction or a null value are classified as non-responders. The Childhood Trauma Questionnaire[28]

will be assessed at baseline as a moderator of treatment response at 16 weeks.

Secondary outcome measures are the Beck Depression Inventory-II,[29] Patient Health Questionnaire,[30] Generalised Anxiety Disorder Assessment,[31] Work and Social Adjustment Scale,[32] EuroQol-five-dimensions-5-level,[33] and cognitive functioning as measured by the THINC-Integrated Tool (THINC-it) (THINC-it Task Force, http://thinc.progress.im/en). Quick Inventory of Depressive Symptomatology[34] is completed for a mechanism study (see online supplementary material appendix 2).

In addition, a purposely designed patient proforma will be used to collect patient NHS service utilisation information at the 16 and 26 weeks follow-up time points. This will cover relevant items listed in the Client Service Receipt Inventory,[35] and tailor the resources items measured following good practice approaches used by the health economics database of instruments for resource use measurement group to estimate costs. Patient acceptability will be measured on a 1–5 scale from unacceptable to acceptable and patient experience of overall improvement will be assessed using the Patient Global Impression of Change (1–5 scale much worse to much better)[36]

**Table 2** Study schedule and assessments

| Outcome measures | Baseline assessment (consent to the study) | Baseline MRI scan | Treatment Monday–Friday for 4–6 weeks | 8 weeks follow-up assessment | 16 weeks follow-up assessment | 16 weeks MRI scan | 26 weeks follow-up assessment |
|---|---|---|---|---|---|---|---|
| Visit window | | Within 14 days of Baseline Assessment | +14 days of MRI Scan | ±1 week from Randomisation | ±1 week from Randomisation | Within 14 days of 16 week Follow-up Assessment | ±1 week from randomisation |
| HDRS-17 | ✓ | | ✓Only if baseline assessment exceeds 4 weeks | ✓ | ✓ | | ✓ |
| MGH | ✓ | | ✓ Only if Baseline assessment exceeds 4 weeks | | | | |
| BDI-II | ✓ | | | ✓ | ✓ | | ✓ |
| PHQ-9 | ✓ | | | ✓ | ✓ | | ✓ |
| WSAS | ✓ | | | ✓ | ✓ | | ✓ |
| GAD7 | ✓ | | | ✓ | ✓ | | ✓ |
| EQ-5D-5L | ✓ | | | ✓ | ✓ | | ✓ |
| THINC-it | ✓ | | | ✓ | ✓ | | ✓ |
| SCID-5 | ✓ | | | | | | |
| CTQ | ✓ | | | | | | |
| QIDS-SR | ✓ | | | ✓ | ✓ | | ✓ |
| Client Resource Questionnaire | ✓ | | | | ✓ | | ✓ |
| Patient Acceptability | | | ✓ | ✓ | ✓ | | ✓ |
| Side effects checklist (adverse events) | | | ✓ | ✓ | | | |
| fMRI | | ✓ | | | | ✓ | |
| MRI | | ✓ | | | | ✓ | |
| rsfMRI | | ✓ | | | | ✓ | |
| MRS | | ✓ | | | | ✓ | |
| Diffusion-weighted imaging | | ✓ | | | | ✓ | |

BDI-II, Beck Depression Inventory-II; CTQ, Childhood Trauma Questionnaire; EQ-5D-5L, EuroQoL-five-Dimensions-5-Level; GAD-7, Generalised Anxiety Disorder-7; HDRS-17, Hamilton Depression Rating Scale-17; MGH, Massachusetts General Hospital; MRS, MR spectroscopy; QIDS-SR, Quick Inventory of Depressive Symptomatology; rsfMRI, resting state functional MRI; SCID-5, Structured Clinical Interview-5; THINC-it, THINC-Integrated Tool; WSAS, Work and Social Adjustment Scale.

after each TMS session and at each follow-up time point. A side effect checklist will be completed after each TMS session and 8 weeks follow-up assessment. We will note any changes in medication or other forms of treatment.

Participants will undergo baseline structural MRI, rsfMRI, diffusion weighted imaging scans to assess structural connections between relevant brain regions, and MRS scans (not in London where only fMRI and structural MRI scans will be performed). Treatment target identification will be analysed centrally in Nottingham and communicated to treatment sites. Using GCA of fMRI scans in each subject, coordinates for the stimulation target within the left DLPFC that shows maximal effective

connectivity with the rAI will be identified. For the MRS scans, voxels will be placed in the DLPFC and ACC, using a similar method to prior work with the MEGA-PRESS sequence for GABA-edited MRS.[22] The MRI scans are repeated at 16 weeks to assess FC, effective connectivity and GABA.

### Randomisation

Randomisation will take place immediately prior to the start of the first treatment session, with participants randomised to either rTMS or cgiTBS treatments. Randomisation will be conducted via a web-based randomisation system (Sealed Envelope, www.sealedevelope.com)

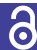

by a named TMS nurse practitioner and healthcare assistant delivering TMS at each centre who will remain un-blinded. Participants will be randomly assigned in a 1:1 ratio into the rTMS and cgiTBS arms using blocks of varying size. Randomisation will be stratified by centre and minimised on severity of depression at baseline and degree of treatment resistance. Baseline depression will be measured by HDRS-17 score and classified as moderate (16-23) or severe (≥24).[37] Treatment resistance will be measured by the MGH TRD staging score and, based on distributions of treatment resistance in the Antiglucocorticoid augmentation of anti-Depressants in Depression (ADD) study,[38] degree of treatment resistance will be classified as low (2–3.5), medium (4-6) and high (≥6.5).

## Statistical analyses
### Efficacy analysis
The primary analysis will test the null hypothesis that treatment with cgiTBS does not increase response rate as measured by HDRS-17 compared with rTMS at 16 weeks on an intent to treat population with any missing response data at 16 weeks being imputed as no response having been achieved. Response at 16 weeks was selected as the primary outcome because of the much larger response rate with cgiTBS than rTMS in our pilot study.[21] A logistic regression will be fitted for the outcome of response with treatment arm, centre, baseline HDRS-17 score and MGH TRD score.

As a secondary analysis a logistic regression model will be repeated in completers (those with 10 or more rTMS or cgiTBS treatments, assessed at baseline and 16 weeks) and a per-protocol analysis. Secondary outcomes are the and proportion of responders and remitters at 8, 16 and 26 weeks, and sustained responders at both 16 and 26 weeks will be compared between groups using logistic regression adjusted for treatment centre, baseline HDRS-17 score and MGH TRD score (5% significance). All outcomes are shown in table 3. Repeated measure linear models will be used for continuous outcomes. Patient acceptability (5-point scale) and safety of both cgiTBS and rTMS (side-effects checklist) will be reported descriptively.

We will explore moderators of depression response at 16 weeks, namely severity of depression by baseline HDRS-17 score, degree of treatment resistance, childhood trauma and age. The number of TMS sessions attended will be explored as a mediator of outcome in exploratory subgroup analyses of the depression response at 16 weeks.

### Economic analysis
The economics analysis will take an NHS and personal social services cost perspective in accordance with National

| Table 3 | Study efficacy outcome measures | |
|---|---|---|
| **Outcomes** | **Measure** | **Time points** |
| Primary Outcome | 17-Item Hamilton Depression Rating Scale (HDRS-17) | Response at 16 weeks (defined as by a 50% or greater reduction in HDRS-17 score from baseline) |
| Secondary Outcomes | HDRS-17 | Mean change in HDRS-17 score compared with baseline at 8, 16 and 26 weeks Response (as defined above) at 8 weeks and at 26 weeks Sustained response at 16 and 26 weeks (defined as a continuing response as defined above following a response at the previous time point) Remitters at 8, 16 and 26 weeks (defined as a score of 7 or less on the HDRS-17) |
| | Beck Depression Inventory-II | At 8, 16 and 26 weeks |
| | Patient Health Questionnaire Quick Inventory of Depressive Symptomatology | At 8, 16 and 26 weeks At 8, 16 and 26 weeks |
| | Generalised Anxiety Disorder Assessment | At 8, 16 and 26 weeks |
| | THINC Integrated Tool | At 8, 16 and 26 weeks |
| | EuroQol-five-Dimensions-five-Level | At 8, 16 and 26 weeks |
| | Work and Social Adjustment Scale | At 8, 16 and 26 weeks |
| | Patient Acceptability (1–5 scale) | After each TMS session and at 8, 16 and 26 weeks |
| | Adverse Events Checklist | After each TMS session and 8 weeks |
| | MRI scan (structural (T1), diffusion-weighted imaging (DWI) for structural connectivity, resting state fMRI for functional connectivity (FC) and effective FC (eFC), and GABA MRS of the DLPFC and insula | T1, DWI, FC, eFC at baseline (all centres) GABA at baseline and T1, DWI, FC, eFC, GABA at 16 weeks (all sites except London). |

DLPFC, dorsolateral prefrontal cortex; fMRI, functional MRI; MRS, MR spectroscopy; TMS, transcranial magnetic stimulation.

Institute of Health and Care Excellence (NICE) guidance, including medication, inpatient and outpatient hospital visits and primary and community care use. It will also take a wider societal perspective to capture the broader effects of rTMS and cgiTBS on depression such as paid employment, caring responsibilities and other implications for friends and family. Resource data will then form the units on which cost data, using sources such as the Unit Cost of Health and Social Care\Personal Social Services Research Unit (PSSRU),[39] the British National Formulary and national reference costs can be attached. The nurse and healthcare assistant at each centre will complete a diary of time spent managing each participant in the RCT to derive treatment costs for rTMS and cgiTBS. The number of treatment sessions for each treatment arm will be carefully recorded and a separate intervention cost assigned to each of the therapies. We will delineate the time spent delivering cgiTBS or rTMS from time spent on research only procedures, such as imaging for delivery of rTMS and additional mechanistic scans in both treatment arms or the extended length of the TBS session for blinding that would not be used in the real clinical world. The outcome measure for the economic evaluation will be the number of quality-adjusted life year (QALY) based on a 6-month time horizon with no discounting for costs or outcomes as they accrue within a 12-month period. An incremental analysis will be used between the two groups and where appropriate an incremental cost effectiveness ratio will be reported between rTMS and cgiTBS. We will use the net monetary benefit framework and implement a net benefit regression,[40] to estimate the extent to which, and the probability that, the cgiTBS intervention is cost-effective compared with standard rTMS at a range of threshold values for the willingness to pay per QALY, generating cost-effectiveness acceptability curves. Data will be analysed for baseline and centre effects. Key cost drivers will be examined using probabilistic sensitivity analysis.

### Qualitative analysis

A purposive sample of 25–30 participants from both arms and all centres, reflecting a mix of demographic characteristics, consent or non-consent to participate, adherence and non-adherence to treatment and follow-up will be selected for qualitative interviews, each lasting for up to an hour, after the 16 weeks assessment (the primary outcome). We will ask about their general views of TMS, benefits from, disadvantages from or dislikes about receiving TMS. Interviews will continue until theme saturation is achieved, that is, no more themes emerge in subsequent interviews. Qualitative interviews of barriers to recruitment, with staff members involved in recruitment, will also be completed to optimise strategies to improve recruitment as the RCT proceeds. All interviews will be recorded and transcribed verbatim. Inductive thematic analysis using a grounded approach will be adopted.

### Sample size

Sample size is calculated on the minimum clinically important difference in responder rates from baseline to 16 weeks of 15% (equivalent to a number needed to treat of 7) between two active treatments (cgiTBS vs rTMS) in favour of the experimental condition cgiTBS. This is an effect size regarded in the literature as a clinically important difference in studies using invasive approaches to manage TRD such as vagal nerve stimulation.[40] The most recent meta-analysis of 16 RCTs in 977 participants with TRD reports a response rate on the HDRS-17 and the Montgomery-Åsberg Depression Rating Scale of 26.5% for rTMS at the end of treatment vs 13.1% for sham TMS.[6] Assuming a response rate on the HDRS-17 of 26.5% with rTMS and 41.5% with cgiTBS, 368 patients in total (184 per arm) will provide a power of 80% to detect the difference in response rates at a 5% significance level (two tailed), allowing for 15% lost to follow-up at 16 weeks.[6]

### Blinding

Patients, referring clinical teams and the outcomes assessors will be kept blind with respect to the treatment protocol assigned and administered. Any unintended unblinding of outcome assessors will be recorded and another assessor will complete all further assessments for that participant. At each assessment, the outcomes assessor will be asked to guess the treatment allocation of the participant.

## ETHICS AND DISSEMINATION
### Ethical considerations

Ethical approval was granted by East Midlands Leicester Central Research Ethics Committee (ref: 18/EM/0232) on 30/08/2018. The study was registered on 02/10/2018 (ISRCTN19674644) under the public title 'BRIGhT-MIND: brain imaging guided transcranial magnetic stimulation in depression'. The first participant was randomised on 22 January 2019. Study finish is scheduled for September 2021. Overview of the study is performed by an external independent Data Monitoring Ethics Committee reporting to a Trial Steering Committee.

Expenses will be covered for participation in the study along with a £10.00 shopping voucher at 16 and 26 weeks follow-up assessments as a mark of respect and gratitude for the time and input of the participants to the follow-up aspects of the trial.

### Safety considerations

Internationally agreed definitions of adverse events (any untoward medical occurrence in a clinical trial subject administered TMS whether or not it has a causal relationship with TMS) and serious adverse events (any adverse event or adverse reaction that results in death, is life-threatening, requires hospitalisation or prolongation of existing hospitalisation, results in persistent or significant disability or incapacity, or is a congenital anomaly or birth defect). All participants will be asked about adverse events after every treatment (immediately

and a maximum of 72 hours later) until they are resolved. Any participant found to be at risk to themselves (suicide, neglect) or others, or developing a serious adverse event will be referred to relevant clinical services. A review by a clinical expert in TRD will be offered to any participant whose depression has become more severe at 16 and 26 weeks for safety reasons.

### Dissemination

All participants will be sent a report summary of the results. Results of the study will be published in peer-reviewed academic journals and communicated widely to academic, profession and public audiences through conferences, media and social media.

## PATIENT AND PUBLIC INVOLVEMENT

Regarding this study a coproduction approach has been taken. We worked with the Involvement Centre at Nottinghamshire Healthcare Trust to create a Magnetic Stimulation Advisory Group who co-produced the treatment pathway for prior pilot work[21] and the current study.

During the study, each centre will be invited to contribute at least two patient and public involvement (PPI) representatives to form a Lived Experience Advisory Panel (LEAP) who will contribute to all study meetings and a representative will be invited to attend all research meetings. At each site PPI representatives will seek opinions, test ideas and gain support for the study. Participant information sheets and other study materials will be codesigned. They will see oversee the interpretation of our findings, particularly the emerging qualitative analysis on barriers to recruitment and patient acceptability of TMS. The PPI process will be overseen by an experienced PPI lead who has trained supported and mentored many PPI groups over recent years. The voice of experts by experience will be heard in all dissemination activities, including presentations or publications.

## DISCUSSION

Although TMS is an established efficacious and safe treatment approach for TRD, research is required to establish if it is possible to increase the proportion of people with TRD who derive a sustained improvement in depression response.[8] Based on theoretical and preliminary empirical evidence, this study explores the potential of using precise personalised neuroanatomical stimulation at a site where affective, cognitive control and default mode networks overlap in neuroanatomical space as determined by MRI derived maximal strength of connectivity between the rAI and the left DLPFC. It will be compared with standard rTMS in a large multicentre RCT. Both rTMS and iTBS are effective against sham treatments[6 8 10] and recommended for clinical practice in the UK.[8] Furthermore, standard iTBS is non-inferior but not superior to rTMS.[12] Thus, if the current RCT showed that cgiTBS was superior in efficacy, it would be reasonable to conclude

that the increased efficacy was due to the personalisation of the site of iTBS treatment.

The design of the trial aims to have high external validity and generalisability to inform routine clinical care. Therefore, inclusion and exclusion criteria are broad and confined to ensuring that safety, efficacy and the investigation of mechanism of action are not compromised. Our definition of TRD[26] is one of the most widely used but it does not consider psychological treatment and underestimates treatment resistance in those who refuse or are not offered antidepressants or ECT in the current episode because of lack of effectiveness or intolerance in previous depression episodes. The broad inclusion and exclusion criteria, large sample size, inclusion of clinical and MRI variables and long duration of follow-up will permit a wide variation in potential predictors and non-predictors of response for clinical practice and further research.

Compromises have been made to the design of the study to improve blinding. cgiTBS and rTMS have been matched procedurally so both use the same MRI and neuronavigation system, although these are not currently being used in clinical practice in UK. TMS treatments have also been matched in terms of pulses received and duration of treatment, although the duration of cgiTBS is delivered over a much longer duration but at a lower motor threshold than the United States Federal Drug Agency protocol used in the Three-D study.[12] We decided to use the motor threshold that was well tolerated in our pilot study.[21] We also did not include a third arm of cgrTMS because our pilot study showed a short duration of response similar to standard rTMS.[21] The use of sham coils to mask the sound and sensation of iTBS and rTMS was explored but effective shams were not available. As a result, entry into the study is restricted to people who had never previously received TMS.

A coproductive approach has been taken with our patient and public LEAP group and this has been particularly important around informing participants accurately about the study and managing participant expectations. By doing this we hope to optimise recruitment and retention to the study.

In many countries, 3 Tesla MRI is increasingly available so that cgiTBS could be implemented into routine care for TRD. Should the RCT demonstrate that cgiTBS is clinically effective at 16 weeks and cost-effective compared with rTMS, then the possibility of maintaining participants with one full course of cgiTBS with booster sessions two or three times per year using the same site of stimulation from a baseline MRI might be feasible. If it is not effective, then other parameters of delivering TMS should be the focus of research and development.

**Author affiliations**
[1]Psychiatry, University of Nottingham, Nottingham, UK
[2]Nottinghamshire Healthcare NHS Foundation Trust, Nottingham, UK
[3]Camden and Islington NHS Foundation Trust, London, UK
[4]Arthritis Research UK Pain Centre, University of Nottingham, Nottingham, UK
[5]Sir Peter Mansfield Imaging Centre, University of Nottingham, Nottingham, UK

[6]University of Leicester, Leicester, UK
[7]Nottingham, UK
[8]University of Newcastle upon Tyne, Newcastle upon Tyne, UK
[9]University of Nottingham, Nottingham, UK
[10]Leicester Clinical Trials Unit, University of Leicester, Leicester, UK
[11]School of Medicine, University of Nottingham, nottingham, UK
[12]Northamptonshire Healthcare NHS Foundation Trust, Kettering, UK
[13]Precision Imaging Beacon, University of Nottingham, Nottingham, UK

**Acknowledgements** We thank the ongoing support of the Funders, the Patient and Public Involvement representatives, participants, research and treatment staff at the four centres, the NIHR Clinical Research Network in East Midlands, North East and North Thames, the clinical, methodological and lay members of the independent Data Monitoring Ethics Committee and Trial Steering Committees, the Leicester Clinical Trials Unit and all other staff whom are involved in the study. RM is supported by the Nottingham National Institute for Health Research (NIHR) Biomedical Research Centre, NIHR MindTech Med Tech and in Vitro Centre and NIHR Applied Research Centre East Midlands.

**Collaborators** BRIGhTMIND study team: Lorraine Bastick, Rosie Carr, Alison Cartlidge, Harry Clark, William Cottam, Robert De Vai, Linda Davison, John Gledhill, Adele Gregory, Christopher Griffiths, Andrew Hamilton, Delilah Harding, Kelly Heath, Rachel Hobson, Gbeminiyi Ireoluwa, Najat Khalifa, Kate Johnstone, Charlotte Kirkland, Mark Liddle, Jessica Lynch, Neil Nixon, Jehill Parikh, Isabel Reid, Noemi Reiner, Sandra Simpson, Beverley Smith, Tina Sore, Joseph Stone, Carly Taylorson, Rebecca Toney, Claire Turner, Sarah Wilkinson, Andy Willis, Tom Willis.

**Contributors** All authors approved the final manuscript. RM is the chief investigator and guarantor of the project. He designed the study, obtained funding and wrote the final draft of the protocol. LW wrote operational guidelines for the project, qualitative analysis protocols and the first draft on this manuscript. MA is the principal investigator for the London site, obtained funding, and designed the TMS schedules. DPA is the neuroimaging lead, led some of the underpinning work designed the neuroimaging and obtained grant funding. SB carried out power calculations for the study and designed the statistical analysis. PB has led the patient and public involvement aspects of the project and their contributions to the operation of the project. AB designed the neuroimaging and obtained grant funding. PMB designed some of the neuroimaging protocol and is analysing the imaging data. CB designed the statistical analysis and obtained funding for the study. SI led some of the underpinning work, designed the neuroimaging and TMS parts of the protocol and obtained funding. MJ designed the health economic analysis, the health economic data collection forms and will be responsible for the health economic management of the trial and oversee all the health economic analysis. CK-H wrote protocols for TMS in the study and carried out some of the neuroimaging analysis. AO-K is the principal investigator for the Northampton site, obtained funding, and designed the TMS schedules. SL, the principal investigator for the Nottingham site, carried out some of the underpinning work, obtained funding, and designed the TMS schedules. PL carried out some of the underpinning work and designed some of the neuroimaging and TMS work, and obtained funding. HM-W is the principal investigator for the Newcastle site, designed the cognitive assessment and assessment of treatment resistance and obtained funding. SPP designed and carried out the computer programming for TMS site location, helped design the Neuronavigation and TMS protocols, and is carrying out neuroimaging analysis. ASDP designed the Statistical Analysis Plan and is carrying out the statistical analysis. LT is the qualitative research lead, designed the qualitative parts of the project and obtained funding. YW is the trial manager of the study and has led the writing of study operating procedures, ethics and governance protocols for the study.

**Funding** This project (project reference 16/44/22) is funded by the Efficacy and Mechanism Evaluation (EME) Programme, an MRC and NIHR partnership. Equipment for the study was provided by Magstim Company. SponsorResearch and Evidence Team, Nottinghamshire Healthcare NHS Foundation Trust. The views expressed in this publication are those of the author(s) and not necessarily those of the MRC, NIHR, Magstim Company or the Department of Health and Social Care.

**Disclaimer** The views expressed in this publication are those of the author(s) and not necessarily those of the MRC, NIHR, Magstim Company or the Department of Health and Social Care.

**Competing interests** AO-K has received fees for consultancy with Magstim.

**Patient and public involvement** Patients and/or the public were involved in the design, or conduct, or reporting, or dissemination plans of this research. Refer to the Methods section for further details.

**Patient consent for publication** Not required.

**Provenance and peer review** Not commissioned; externally peer reviewed.

**ORCID iD**
Richard Morriss http://orcid.org/0000-0003-2910-4121

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
