## [Reviewer comments · BMJ Open]

ARTICLE DETAILS

TITLE (PROVISIONAL)	Connectivity guided theta burst transcranial magnetic stimulation versus repetitive transcranial magnetic stimulation for treatment resistant moderate to severe depression: study protocol for a randomised double-blind controlled trial (BRIGHtMIND).
AUTHORS	Morriss, Richard; Webster, Lucy; Abdelghani, Mohamed; Auer, Dorothee; Barber, Shaun; Bates, Peter; Blamire, Andrew; Briley, Paul; Brookes, Cassandra; Iwabuchi, Sarina; James, Marilyn; Kaylor-Hughes, Catherine; Lankappa, Sudheer; Liddle, Peter; McAllister-Williams, Hamish; O'Neill-Kerr, Alex; Pszczolkowski Parraguez, Stefan; Suazo Di Paola, Ana; Thomson, Louise; Walters, Yvette

VERSION 1 - REVIEW

REVIEWER	Dr. Yuliya Knyahnytska Centre for Addiction and Mental Health, University of Toronto, Toronto, Canada
REVIEW RETURNED	28-Mar-2020

GENERAL COMMENTS	This is very well written manuscript. It is well organized, logical, and information presented is clinically relevant. It was a pleasure reading it and a great distraction at the midst of pandemic. I honestly have nothing to add here, just want to wish authors good luck with publication.
--

REVIEWER	Tianmei Si Peking University Institute of Mental Health, Beijing, China
REVIEW RETURNED	10-Apr-2020

GENERAL COMMENTS	This manuscript provides a study protocol clearly. And the manuscript is organized well. Only one minor concern for this protocol, there are two assessment of mood used in this manuscript, HAMD and MADRS depression scale, which one is the primary outcome? why author use two scales in this study?
--

REVIEWER	Phern-Chern Tor Institute of Mental Health (Singapore)
REVIEW RETURNED	11-Apr-2020

GENERAL COMMENTS	Dear Authors Thank you for the manuscript on an ambitious multicentre study comparing different TMS modalities in the treatment of treatment resistant depression. The study methodology is robust. I do have some suggestions to elaborate on some design choices that would help readers. 1) Page 6 Line 52 states "standard rTMS" as a comparator but the methods detail neuronavigation for the rTMS comparator. In many routine clinical settings standard rTMS does not include neuronavigation. I would suggest clarifying that the comparator is not standard rTMS but neuronavigated rTMS 2) Page 8 line 17 talks about exclusion of patients with recent (2 week) prescription of AEDs. Please clarify if AED prescription prior to 2 weeks before baseline assessment is allowed 3) Page 9 line 14: Why is the cgiTBS dosage only 80% of RMT when the THREE-D study this study references at 120% RMT and the active comparator 120% RMT standard 10Hz rTMS? Will this not underdose the investigational treatment? If there is a logic for this the average reader would appreciate some clarification. In addition the desire to make the 2 TMS treatments comparable in time is laudable but spreading out a 3 minute treatment over 30 minutes (if I understand the protocol on page 9 line 13 to 26 correctly) may be a very different treatment from iTBS over 3 minutes. Again a justification of how this novel iTBS treatment which is different from the US FDA approved iTBS regime would be useful.
---

REVIEWER	Frank P MacMaster and Kayla Stone (PDF) University of Calgary, Canada
REVIEW RETURNED	15-Apr-2020

GENERAL COMMENTS	TO THE AUTHORS Thank you for the opportunity to read your manuscript. Below is what I hope is constructive feedback for improvement or consideration. SUMMARY Morriss and colleagues plan to conduct a randomized double-blind controlled trial comparing the clinical efficacy of connectivity-guided intermittent theta burst stimulation and repetitive transcranial magnetic stimulation for adults with treatment resistant depression. As a secondary outcome, they will also examine the effects on mood, cognition, psychosocial functioning, quality of life, and overall cost effectiveness, patient acceptability, safety, mechanism of action and predictors of response. STRENGTHS This trial protocol is thorough and clear. It addresses many important questions in the field about treatment resistant depression right now,
--

using a multicenter approach.

COMMENTS/CONCERNS/QUERIES/CLARIFICATIONS

Abstract

1. The authors state where participants will be recruited from, but where will the data be collected? (as per CONSORT guidelines)
2. Source of funding is not listed in abstract (as per CONSORT guidelines)

Introduction

1. Page 3, lines 28-45: rTMS and TBS are not well-defined. I would consider adding a sentence to clarify what they are/how they work.
2. Page 3, line 51: "...found unilateral intermittent TBS (iTBS) and bilateral iTBS to be robustly effective..."
 - a. Please clarify: iTBS administered to which part of the brain?
3. Page 4, lines 16-17: "...connectivity of the insula predicts efficacy of cognitive behaviour therapy and antidepressant treatment in patients with depression."
 - a. Please clarify: Do the authors mean connectivity to the left DLPFC? And in which direction (increased or decreased connectivity predicts efficacy?)
4. Page 4, line 18: Do the authors mean one biotype of MDD?
5. Page 4, lines 34-43: Perhaps the authors could add a sentence at the end of this paragraph to clarify that these measures will also be examined in the current study. Otherwise, the goal of this paragraph is unclear.
6. Page 4, line 4: Objective is stated, but my concern here is that cgtTBS might be better compared to cgrTMS (with standard rTMS as a 3rd arm). If not addressing in study design, I invite the authors to address this in the discussion.
7. Other, minor: reference locations are inconsistent throughout (i.e. sometime before period/comma, sometimes after). Page 3: remove comma before reference [5], line 30 and before reference [10], line 50.

Methods

1. Table 1, under exclusion criteria "tattoo's" should be "tattoos"
2. Table 1, under exclusion criteria "at risk of suicidality" – How will this be assessed?
3. Figure 1: rTMS treatment box, "session" should be plural
4. Page 7, line 3-4: "Participants in both treatment arms will receive neuronavigation guided single coil stimulation for 20 TMS sessions delivered over 4-6 weeks"
 - a. Please clarify: Neuronavigated to which part of the brain?
5. Page 7, line 10: "There must not be a gap of more than 4 days between treatments in the course of 20."
 - a. Please clarify: What is the rationale for choosing 4 days?
6. Page 7, line 40: Hypothesis is stated, but could the authors clarify why the primary objective only focuses on 16 weeks? (e.g. and not 8, or 26 weeks).
7. Page 12, line 16: Secondary voxel location for MRS is ACC, but why not insula (too)?

Discussion

1. A possible limitation should be mentioned. The authors do not plan to compare cgtTBS to cg-rTMS (rather, just standard TMS). There is evidence to suggest that cg-rTMS is superior to standard TMS method, albeit the focus was on the connectivity between left DLPFC and subgenual cingulate (Fox et al., 2012, Biological Psychiatry). It would be useful to have a third (even exploratory)

	arm. Other 1. Appendix 1, line 53, the word 'dose' is missing the 'e' TO THE EDITOR Thank you for the opportunity to review for the journal. Attached is what I hope is constructive feedback for improvement or consideration. Overall, the manuscript is a well-written and the protocol is a feasible approach to examining the efficacy of connectivity-guided (cg) iTBS for treatment resistant depression. Most of my comments are related to clarifying the introduction and methodological details. However, my concern is that there is no 'connectivity-guided rTMS' treatment. Rather, the authors choose to compare cfiTBS to standard rTMS. The rationale for choosing standard rTMS is clear (i.e. because it is currently the standard rTMS treatment for treatment resistant depression). However, a third (cgrTMS) arm would have enhanced the trial. Since the first participant was already randomized as of January 2019, a note about this in the discussion would suffice. I would endorse this manuscript for publication following these revisions.
--	--

VERSION 1 – AUTHOR RESPONSE

Reply to Reviewer Comments.

Reviewer 1

No comments.

Reviewer 2

There are two assessment of mood used in this manuscript, HAMD and MADRS depression scale, which one is the primary outcome? Why author use two scales in this study?

Response. There is only one interview measure of depression, the 17-item Hamilton Depression Rating scale. We are using 2 self-rated measures of depression in the full trial, the PHQ-9 since this is routinely used by the NHS in the UK, and the Beck Depression Inventory since this was related to changes in functional connectivity and GABA to glutamate complex in our pilot study. QIDS is being used in a sample of participants for a substudy. The MADRS is not being used.

Reviewer 3

1) Page 6 Line 52 states "standard rTMS" as a comparator but the methods detail neuronavigation for the rTMS comparator. In many routine clinical settings standard rTMS does not include neuronavigation. I would suggest clarifying that the comparator is not standard rTMS but neuronavigated rTMS.

Response. We have clarified this point in the protocol as the referee suggests (see page 5 revised manuscript, paragraph 1 lines 1 and 2, and paragraph 2 line 2).

2) Page 8 line 17 talks about exclusion of patients with recent (2 week) prescription of AEDs. Please clarify if AED prescription prior to 2 weeks before baseline assessment is allowed.

Response. We have clarified this point as the referee suggests. AEDs were allowed prior to the 2 week period before baseline (see Table 1 after pregabalin, gabapentin and lamotrigine, and after benzodiazepines and hypnotic drugs).

3) Page 9 line 14: Why is the cgiTBS dosage only 80% of RMT when the THREE-D study this study references at 120% RMT and the active comparator 120% RMT standard 10Hz rTMS? Will this not underdose the investigational treatment? If there is a logic for this the average reader would appreciate some clarification. In addition the desire to make the 2 TMS treatments comparable in time is laudable but spreading out a 3 minute treatment over 30 minutes (if I understand the protocol on page 9 line 13 to 26 correctly) may be a very different treatment from iTBS over 3 minutes. Again a justification of how this novel iTBS treatment which is different from the US FDA approved iTBS regime would be useful.

Response. We utilised the same treatment using cgiTBS that we utilised in the pilot study which showed some preliminary evidence of prolonged duration of depression response on the HAMD, BDI as well as changes in functional connectivity and GABA/glutamate complex. We planned the study and obtained funding it before the 3D study and FDA guidance for iTBS were released. We considered this data but felt that increasing the RMT to 120% might reduce the tolerability of iTBS. We have added a sentence to our discussion highlighting that the cgiTBS is different from FDA approved iTBS in both duration and RMT threshold (see discussion page 16, paragraph 3 lines 4 to 7)

Reviewer 4

Abstract

1. The authors state where participants will be recruited from, but where will the data be collected? (as per CONSORT guidelines)

Response. We have reviewed abstracts of the last 10 published BMJ Open protocols of randomised controlled trials. This information is not supplied in any of these abstracts. We have therefore not added the information.

2. Source of funding is not listed in abstract (as per CONSORT guidelines)

Response. We have reviewed abstracts of the last 10 published BMJ Open protocols of randomised controlled trials. This information is not supplied in any of these abstracts. We have therefore not added the information.

Introduction

1. Page 3, lines 28-45: rTMS and TBS are not well-defined. I would consider adding a sentence to clarify what they are/how they work.

Response. We have rewritten the definitions of rTMS and TBS as recommended. See page 3 2nd paragraph lines 3-5 for rTMS and page 3, paragraph 4, lines 2to 3.

2. Page 3, line 51: "...found unilateral intermittent TBS (iTBS) and bilateral iTBS to be robustly effective..."

a. Please clarify: iTBS administered to which part of the brain?

Response. We have added the information requested. See page 3, paragraph 4, lines 7to 8.

3. Page 4, lines 16-17: "...connectivity of the insula predicts efficacy of cognitive behaviour therapy and antidepressant treatment in patients with depression."

a. Please clarify: Do the authors mean connectivity to the left DLPFC? And in which direction (increased or decreased connectivity predicts efficacy?)

Response. We have clarified the findings of this study. See page 4, paragraph 2, lines 6 and 7.

4. Page 4, line 18: Do the authors mean one biotype of MDD?

Response. We have added MDD to biotype to address this point. See page 4, paragraph 2, line 8.

5. Page 4, lines 34-43: Perhaps the authors could add a sentence at the end of this paragraph to clarify that these measures will also be examined in the current study. Otherwise, the goal of this paragraph is unclear.

Response. The purpose of this paragraph is to explain the rationale for the study and why we might expect to see differences in response between the two treatment arms other than the results of our small pilot study. We intend to explore a number of imaging mechanisms of which these are some but we plan to publish a separate imaging protocol because there is not the space to give full details of this particular aspect of the work.

6. Page 4, line 4: Objective is stated, but my concern here is that cgrTBS might be better compared to cgrTMS (with standard rTMS as a 3rd arm). If not addressing in study design, I invite the authors to address this in the discussion.

Response. We are not conducting a three arm RCT because the sample size would be prohibitively large. This is already one of the largest TMS RCTs ever undertaken. We did not include a cgrTMS arm because our pilot data suggested that there were not changes in the duration of response and fewer improvements in connectivity and GABA/glutamate complex with cgrTMS. Theoretically iTBS produces long-term potentiation at more distal sites so our pilot study results coincide with theoretical effects. We have added a sentence to this effect in the discussion. See page 16, paragraph 3, lines 7-8.

7. Other, minor: reference locations are inconsistent throughout (i.e. sometime before period/comma, sometimes after). Page 3: remove comma before reference [5], line 30 and before reference [10], line 50.

Response. These are corrected.

Methods

1. Table 1, under exclusion criteria "tatoo's" should be "tattoos"

Response. This spelling mistake has been corrected. See Table 1.

2. Table 1, under exclusion criteria "at risk of suicidality" – How will this be assessed?

Response. This will be assessed clinically on the basis of the SCID and scores on suicide items on the HDRS-17, PHQ-9 and BDI. Participants showing possible suicidality are discussed with the Principal Investigator at each site. We have not added this additional detail since this is common practice but can do if the editors wish us to.

3. Figure 1: rTMS treatment box, "session" should be plural

Response This change has been made to he figure.

4. Page 7, line 3-4: "Participants in both treatment arms will receive neuronavigation guided single coil stimulation for 20 TMS sessions delivered over 4-6 weeks"

a. Please clarify: Neuronavigated to which part of the brain?

Response. This is explained in turn for each treatment arm in the following 2 paragraphs.

5. Page 7, line 10: "There must not be a gap of more than 4 days between treatments in the course of 20."

a. Please clarify: What is the rationale for choosing 4 days?

Response. Short courses of TMS or TMS delivered infrequently are thought to be less effective in the treatment of depression. Public holidays in the UK might lead to no TMS treatment for 3 days. We therefore allowed some flexibility in the delivery of the course of treatment of 20 session of TMS by allowing it to be delivery over 6 weeks but with no more than 4 days between treatments to ensure that treatment courses were completed in a timely manner that is thought to be effective.

6. Page 7, line 40: Hypothesis is stated, but could the authors clarify why the primary objective only focuses on 16 weeks? (e.g. and not 8, or 26 weeks).

Response. We needed to select on time point as a primary objective according to the statistical and trial design advice we received and our funders insisted upon. In our pilot work there was an important difference in outcome at 3 months after treatment was completed not at one month after treatment was completed. We did not have pilot data at 26 weeks so we did not use this time point as the primary outcome. We have added a sentence to explain why 16 weeks is the primary outcome. See page 11, paragraph 1, lines 4 to 5.

7. Page 12, line 16: Secondary voxel location for MRS is ACC, but why not insula (too)?

Response. We will examine the insula and have added this to Table 3 on page 12.

Discussion

1. A possible limitation should be mentioned. The authors do not plan to compare cgtTBS to cg-rTMS (rather, just standard TMS). There is evidence to suggest that cg-rTMS is superior to standard TMS method, albeit the focus was on the connectivity between left DLFPC and subgenual cingulate (Fox et al., 2012, Biological Psychiatry). It would be useful to have a third (even exploratory) arm.

Response. We covered this point in the introduction point 6. And as stated previously we have added a sentence in the discussion to address this point. As stated under this point we have added a sentence to the discussion to address this point. Our pilot work did not find that cg-rTMS produced results that were any different to what we could achieve with standard rTMS

Other

1. Appendix 1, line 53, the word 'dose' is missing the 'e'

Response. This typographical error has been corrected.

VERSION 2 – REVIEW

REVIEWER	Phern-Chern Tor Institute of Mental Health Singapore
REVIEW RETURNED	07-May-2020

GENERAL COMMENTS	Thank you my queries have been addressed
--

REVIEWER	Frank MacMaster University of Calgary, Canada
REVIEW RETURNED	06-May-2020

GENERAL COMMENTS	All previous concerns addressed sufficiently.
---